# A Robust Fuel Optimization Strategy For Hybrid Electric Vehicles: A Deep Reinforcement Learning Based Continuous Time Design Approach

## Abstract

This paper deals with the fuel optimization problem for hybrid electric vehicles in reinforcement learning framework. Firstly, considering the hybrid electric vehicle as a completely observable non-linear system with uncertain dynamics, we solve an open-loop deterministic optimization problem to determine a nominal optimal state. This is followed by the design of a deep reinforcement learning based optimal controller for the non-linear system using concurrent learning based system identifier such that the actual states and the control policy are able to track the optimal state and optimal policy, autonomously even in the presence of external disturbances, modeling errors, uncertainties and noise and signigicantly reducing the computational complexity at the same time, which is in sharp contrast to the conventional methods like PID and Model Predictive Control (MPC) as well as traditional RL approaches like ADP, DDP and DQN that mostly depend on a set of pre-defined rules and provide sub-optimal solutions under similar conditions. The low value of the H-infinity ($H_\infty$) performance index of the proposed optimization algorithm addresses the robustness issue. The optimization technique thus proposed is compared with the traditional fuel optimization strategies for hybrid electric vehicles to illustate the efficacy of the proposed method.

## 1 Introduction

Hybrid electric vehicles powered by fuel cells and batteries have attracted great enthusiasm in modern days as they have the potential to eliminate emissions from the transport sector. Now, both the fuel cells and batteries have got several operational challenges which make the separate use of each of them in automotive systems quite impractical. HEVs and PHEVs powered by conventional diesel engines and batteries merely reduce the emissions, but cannot eliminate completely. Some of the drawbacks include carbon emission causing environmental pollution from fuel cells and long charging times, limited driving distance per charge, non-availability of charging stations along the driving distance for the batteries. Fuel Cell powered Hybrid Electric Vehicles (FCHEVs) powered by fuel cells and batteries offer emission-free operation while overcoming the limitations of driving distance per charge and long charging times. So, FCHEVs have gained significant attention in recent years. As we find, most of the existing research which studied and developed several types of Fuel and Energy Management Systems (FEMS) for transport applications include Sulaiman et al. (2018) who has presented a critical review of different energy and fuel management strategies for FCHEVs. Li et al. (2017) has presented an extensive review of FMS objectives and strategies for FCHEVs. These strategies, however can be divided into two groups, i.e., model-based and model-free. The model-based methods mostly depend on the discretization of the state space and therefore suffers from the inherent curse of dimensionality. The coumputational complexity increases in an exponential fashion with the increase in the dimension of the state space. This is quite evident in the methods like state-based EMS (Jan et al., 2014; Zadeh et al., 2014; 2016), rule-based fuzzy logic strategy (Motapon et al., 2014), classical PI and PID strategies (Segura et al., 2012), Potryagin's minimum principle (PMP) (Zheng et al., 2013; 2014), model predictive control (MPC) (Kim et al., 2007; Torreglosa et al., 2014) and differential dynamic programming (DDP) (Kim et al., 2007). Out of all these methods, differential dynamic programming is considered to be computationally quite

efficient which rely on the linearization of the non-linear system equations about a nominal state trajectory followed by a policy iteration to improve the policy. In this approach, the control policy for fuel optimization is used to compute the optimal trajectory and the policy is updated until the convergence is achieved.

The model-free methods mostly deal with the Adaptive Dynamic Programming (Bithmead et al., 1991; Zhong et al., 2014) and Reinforcement Learning (RL) based strategies (Mitrovic et al., 2010; Khan et al., 2012) icluding DDP (Mayne et al., 1970). Here, they tend to compute the control policy for fuel optimization by continous engagement with the environment and measuring the system response thus enabling it to achieve at a solution of the DP equation recursively in an online fashion. In deep reinforcement learning, multi-layer neural networks are used to represent the learning function using a non-linear parameterized approximation form. Although a compact paremeterized form do exist for the learning function, the inability to know it apriori renders the method suffer from the curse of dimensionality ($O(d^2)$ where, $d$ is the dimension of the state space), thus making it infeasible to apply to a high-dimemsional fuel managememt system.

The problem of computational complexity of the traditional RL methods like policy iteration (PI) and value iteration (VI) (Bellman et al., 1954; 2003; Barto et al., 1983; Bartsekas, 2007) can be overcome by a simulation based approach (Sutton et al., 1998) where the policy or the value function can be parameterized with sufficient accuracy using a small number of parameters. Thus, we will be able to transform the optimal control problem to an approximation problem in the parameter space (Bartesekas et al., 1996; Tsitsiklis et al., 2003; Konda et al., 2004) side stepping the need for model knowledge and excessive computations. However, the convergence requires sufficient exploration of the state-action space and the optimality of the obtained policy depends primarily on the accuracy of the parameterization scheme.

As a result, a good approximation of the value function is of utmost importance to the stability of the closed-loop system and it requires convergence of the unknown parameters to their optimal values. Hence, this sufficient exploration condition manifests itself as a persistence of excitation (PE) condition when RL is implemented online (Mehta et al., 2009; Bhasin et al., 2013; Vrabie, 2010) which is impossible to be guaranteed a priori.

Most of the traditional approaches for fuel optimization are unable to adrress the robustness issue. The methods described in the literature including those of PID (Segura et al.,2012), Model Predictive Control (MPC) (Kim et al.,2007;Torreglosa et al., 2014) and Adaptive Dynamic Programming (Bithmead et al.,1991;Zhong et al., 2014) as well as the simulation based RL strategies (Bartesekas et al., 1996; Tsitsiklis et al., 2003; Konda et al., 2004 ) suffer from the drawback of providing a sub-optimal solution in the presence of external disturbances and noise. As a result, application of these methods for fuel optimization for hybrid electric vehicles that are plagued by various disturbances in the form of sudden charge and fuel depletion, change in the environment and in the values of the parameters like remaining useful life, internal resistance, voltage and temperature of the battery, are quite impractical.

The fuel optimization problem for the hybrid electric vehicle therefore have been formulated as a fully observed stochastic Markov Decision Process (MDP). Instead of using Trajectory-optimized LQG (T-LQG) or Model Predictive Control (MPC) to provide a sub-optimal solution in the presence of disturbances and noice, we propose a deep reinforcement learning-based optimization strategy using concurrent learning (CL) that uses the state-derivative-action-reward tuples to present a robust optimal solution. The convergence of the weight estimates of the policy and the value function to their optimal values justifies our claim. The two major contributions of the proposed approch can be therefore be summarized as follows:

1) The popular methods in RL literature including policy iteration and value iteration suffers from the curse of dimensionality owing to the use of a simulation based technique which requires sufficient exploration of the state space (PE condition). Therefore, the proposed model-based RL scheme aims to relax the PE condition by using a concurrent learning (CL)-based system identifier to reduce the computational complexity. Generally, an estimate of the true controller designed using the CL-based method introduces an approximate estimation error which makes the stability analysis of the system quite intractable. The proposed method, however, has been able to establish the stability of the closed-loop system by introducing the estimation error and analyzing the augmented system trajectory obtained under the influnece of the control signal.

2) The proposed optimization algorithm implemented for fuel management in hybrid electric vehicles will nullify the limitations of the conventional fuel management approaches (PID, Model Predictive Control, ECMS, PMP) and traditional RL approaches (Adaptive Dynamic Proagramming, DDP, DQN), all of which suffers from the problem of sub-optimal behaviour in the presence of external disturbances, model-uncertainties, frequent charging and discharging, change of enviroment and other noises. The H-infinity ($H_\infty$) performance index defined as the ratio of the disturbance to the control energy has been established for the RL based optimization technique and compared with the traditional strategies to address the robustness issue of the proposed design scheme.

The rest of the paper is organised as follows: Section 2 presents the problem formulation including the open-loop optimization and reinforcement learning-based optimal controller design which have been described in subsections 2.1 and 2.2 respectively. The parametric system identification and value function approximation have been detailed in subsections 2.2.1 and 2.2.2. This is followed by the stability and robustness analysis (using the H-infinity ($H_\infty$) performance index ) of the closed loop system in subsection 2.2.4. Section 3 provides the simulation results and discussion followed by the conclusion in Section 4.

## 2   PROBLEM FORMULATION

Considering the fuel management system of a hybrid electric vehicle as a continous time affine non-linear dynamical system:

$$\dot{x} = f(x, w) + g(x)u,$$
$$y = h(x, v) \tag{1}$$

where, $x \in \mathbb{R}^{n_x}$, $y \in \mathbb{R}^{n_y}$, $u \in \mathbb{R}^{n_u}$ are the state, output and the control vectors respectively, *f(.)* denotes the drift dynamics and *g(.)* denotes the control effectivenss matrix. The functions *f* and *h* are assumed to be locally Lipschitz continuous functions such that *f(0) = 0* and $\nabla f(x)$ is continous for every bounded $x \in \mathbb{R}^{n_x}$. The process noise *w* and measurement noise *v* are assumed to be zero-mean, uncorrelated Gausssian white noise with covariances *W* and *V*, respectively.

*Assumption 1:* We consider the system to be fully observed:

$$y = h(x, v) = x \tag{2}$$

*Remark 1:* This assumption is considered to provide a tractable formulation of the fuel management problem to side step the need for a complex treatment which is required when a stochastic control problem is treated as partially observed MDP (POMDP).

***Optimal Control Problem:*** For a continous time system with unknown nonlinear dynamics *f(.)*, we need to find an optimal control policy $\pi_t$ in a finite time horizon $[0, t]$ where $\pi_t$ is the control policy at time *t* such that $\pi_t = u(t)$ to minimize the cost function given by $J = \int_0^t (x^T Q x + u R u^T) dt + x^T F x$ where, *Q,F > 0* and *R ≥ 0*.

### 2.1   OPEN LOOP OPTIMIZATION

Considering a noise-free non-linear stochastic dynamical system with unknown dynamics:

$$\dot{x} = f(x, 0) + g(x)u,$$
$$y = h(x, v) = x \tag{3}$$

where, $x_0 \in \mathbb{R}^{n_x}$, $y \in \mathbb{R}^{n_y}$, $u \in \mathbb{R}^{n_u}$ are the initial state, output and the control vectors respectively, *f(.)* have their usual meanings and the corresponding cost function is given by $J_d (x_0, u_t) = \int_0^t (x^T Q x + u R u^T) dt + x^T F x$.

*Remark:* We have used piecewise convex function to approximate the non-convex fuel function globally which has been used to formulate the cost function for the fuel optimization.

The open loop optimization problem is to find the control sequence $u_t$ such that for a given initial state $x_0$,

$$\bar{u}_t = arg\ min\ J_d(x_0, u_t),$$
$$subject\ to\ \dot{x} = f(x, 0) + g(x)u, \tag{4}$$
$$y = h(x, v) = x.$$

The problem is solved using the gradient descent approach (Bryson et al., 1962; Gosavi et al., 2003), and the procedure is illustrated as follows:

Starting from a random initial value of the control sequence $U^{(0)} = [u_t^{(0)}]$ the control policy is updated iteratively as

$$U^{(n+1)} = U^{(n)} - \alpha \nabla_{\boldsymbol{U}} J_d(x_0, U^{(n)}), \tag{5}$$

until the convergence is achieved upto a certain degree of accuracy where $U^{(n)}$ denotes the control value at the $n^{th}$ iteration and $\alpha$ is the step size parameter. The gradient vector is given by:

$$\nabla_{\boldsymbol{U}} J_d(x_0, U^{(n)}) = (\frac{\partial J_d}{\partial u_0}, \frac{\partial J_d}{\partial u_1}, \frac{\partial J_d}{\partial u_2}, ....., \frac{\partial J_d}{\partial u_t})|_{(x_0, u_t)} \tag{6}$$

The ***Gradient Descent Algorithm*** showing the approach has been detailed in the **Appendix A.1**.

*Remark 2:* The open loop optimization problem is thus solved using the gradient descent approach considering a black-box model of the underlying system dynamics using a sequence of input-output tests without having the perfect knowlegde about the non-linearities in the model at the time of the design. This method proves to be a very simple and useful strategy for implementation in case of complex dynamical systems with complicated cost-to-go functions and suitable for parallelization.

## 2.2 REINFORCEMENT LEARNING BASED OPTIMAL CONTROLLER DESIGN

Considering the affine non-linear dynamical system given by equation (1), our objective is to design a control law to track the optimal time-varying trajectory $\bar{x}(t) \in \mathbb{R}^{n_x}$. A novel cost function is formulated in terms of the tracking error defined by $e = x(t) - \bar{x}(t)$ and the control error defined by the difference between the actual control signal and the desired optimal control signal. This formulation helps to overcome the challenge of the infinte cost posed by the cost function when it is defined in terms of the tarcking error *e(t)* and the actual control signal signal *u(t)* only (Zhang et al., 2011; Kamalapurkar et al., 2015). The following assumptions is made to determine the desired steady state control.

*Assumption 2:* (Kamalapurkar et al., 2015) The function $g(x)$ in equation (1) is bounded, the matrix $g(x)$ has full column rank for all $x(t) \in \mathbb{R}^{n_x}$ and the function $g^+ : \mathbb{R}^n \rightarrow \mathbb{R}^{mXn}$ which is defined as $g^+ = (g^T g)^{-1}$ is bounded and locally Lipschitz.

*Assumption 3:* (Kamalapurkar et al., 2015) The optimal trajectory is bounded by a known positive constant $b \in \mathbb{R}$ such that $\|\bar{x}\| \leq b$ and there exists a locally Lipschitz function $h_d$ such that $\dot{\bar{x}} = h_d$ $(\bar{x})$ and $g(\bar{x})\ g^+(\bar{x})(h_d(\bar{x}) - f(\bar{x})) = h_d(\bar{x}) - f(\bar{x})$.

Using the *Assumption 2* and *Assumption 3*, the control signal $u_d$ required to track the desired trajectory $\bar{x}(t)$ is given as $u_d(\bar{x}) = g_d^+(h_d(\bar{x}) - f_d)$ where $f_d = f(\bar{x})$ and $g_d^+ = g^+(\bar{x})$. The control error is given by $\mu = u(t) - u_d(\bar{x})$. The system dynamics can now be expressed as

$$\dot{\zeta} = F(\zeta) + G(\zeta)\mu \tag{7}$$

where, the merged state $\zeta(t) \in \mathbb{R}^{2n}$ is given by $\zeta(t) = [e^T, \bar{x}^T]^T$ and the functions $F(\zeta)$ and $G(\zeta)$ are defined as $F(\zeta) = [f^T(e + \bar{x}) - h_d^T + u_d^T(\bar{x})g^T(e + \bar{x}), h_d^T]^T$ and $G(\zeta) = [g^T(e + \bar{x}), \boldsymbol{0}_{mXn}]^T$ where, $\boldsymbol{0}_{mXn}$ denotes a matrix of zeroes. The control error $\mu$ is treated hereafter as the design variable. The control objective is to solve a finite-horizon optimal tracking problem online, i.e., to design a control signal $\mu$ that will minimize the cost-to-go function, while tracking the desired trajectory, is given by $J(\zeta, \mu) = \int_0^t r(\zeta(\tau), \mu(\tau))d\tau$ where, the local cost $r : \mathbb{R}^{2n}XR^m \rightarrow \mathbb{R}$ is given as $r(\zeta, \tau) = Q(e) + \mu^T R\mu$, $R \in \mathbb{R}^{mXm}$ is a positive definite symmetric matrix and $Q : \mathbb{R}^n \rightarrow \mathbb{R}$ is a continous positive definite function.

Based on the assumption of the existence of an optimal policy, it can be characterized in terms of the value function $V^* : \mathbb{R}^{2n} \rightarrow \mathbb{R}$ which is defined as $V^*(\zeta) =$

$min_{\mu(\tau)\epsilon U|\tau\epsilon \mathbb{R}_{t\geq 0}} \int_0^t r(\phi^u(\pi,t,\zeta),\mu(\tau))d\tau$, where $U \in \mathbb{R}^m$ is the action space and $\phi^u(t;t_0,\zeta_0)$ is the trajectory of the system defined by equation (10) with the control effort $\mu : \mathbb{R}_{\geq 0} \to \mathbb{R}^m$ with the initial condition $\zeta_0 \in \mathbb{R}^{2n}$ and the initial time $t_0 \in \mathbb{R}_{\geq 0}$. Taking into consideration that an optimal policy exists and that $V^*$ is continously differentiable everywhere, the closed-form solution (Kirk, 2004) is given as $\mu^*(\zeta) = $ -1/2 $R^{-1}G^T(\zeta)(\nabla_\zeta V^*(\zeta))^T$ where, $\nabla_\zeta(.) = \dfrac{\partial(.)}{\partial x}$. This satisfies the Hamilton-Jacobi-Bellman (HJB) equation (Kirk, 2004) given as

$$\nabla_\zeta V^*(\zeta)(F(\zeta) + G(\zeta)\mu^*(\zeta)) + \bar{Q}(\zeta) + \mu^{*T}(\zeta)R\mu^*(\zeta) = 0 \qquad (8)$$

where, the initial condition $V^* = 0$, and the funtion $\bar{Q} : \mathbb{R}^{2n} \to \mathbb{R}$ is defined as $\bar{Q}([e^T, \hat{x}^T]^T) = Q(e)$ where, $(e(t), \hat{x}(t)) \in \mathbb{R}^n$.

Since, a closed-form solution of the HJB equation is generally infeasible to obtain, we sought an approximate solution. Therefore, an actor-critic based method is used to obtain the parametric estimates of the optimal value function and the optimal policy which are given as $\hat{V}(\zeta, \hat{W}_c)$ and $\hat{\mu}(\zeta, \hat{W}_a)$ where, $\hat{W}_c \in \mathbb{R}^L$ and $\hat{W}_a \in \mathbb{R}^L$ define the vector paramater estimates. The task of the actor and critic is to learn the corresponding parameters. Replacing the estimates $\hat{V}$ and $\hat{\mu}$ for $V^*$ and $\hat{\mu}^*$ in the HJB equation, we obtain the residual error, also known as the Bell Error (BE) as $\delta(\zeta, \hat{W}_c, \hat{W}_a) = \bar{Q}(\zeta) + \hat{\mu}^T(\zeta, \hat{W}_a)R\hat{\mu}(\zeta, \hat{W}_a) + \nabla_\zeta \hat{V}(\zeta, \hat{W}_c)(F(\zeta) + G(\zeta)\hat{\mu}(\zeta, \hat{W}_a))$ where, $\delta : \mathbb{R}^{2n}$ X $\mathbb{R}^L$ X $\mathbb{R}^L \to \mathbb{R}$. The solution of the problem requires the actor and the critic to find a set of parameters $\hat{W}_a$ and $\hat{W}_c$ respectively such that $\delta(\zeta, \hat{W}_c, \hat{W}_a) = 0$ and $\hat{\mu}^T(\zeta, \hat{W}_a) = $ -1/2 $R^{-1}G^T(\zeta)(\nabla_\zeta V^*(\zeta))^T$ where, $\forall \zeta \in \mathbb{R}^n$. As the exact basis fucntion for the approximation is not known apriori, we seek to find a set of approximate parameters that minimizes the BE. However, an uniform approximation of the value function and the optimal control policy over the entire operating domain requires to find parameters that will able to minimize the error $E_s : \mathbb{R}^L$ X $\mathbb{R}^L \to \mathbb{R}$ defined as $E_s(\hat{W}_c, \hat{W}_a) = sup_\zeta(|\delta, \hat{W}_c, \hat{W}_a|)$ thus, making it necessary to have an exact knowledge of the system model. Two of the most popular methods used to render the design of the control strategy robust to system uncertainties in this context are integral RL (Lewis et al., 2012; Modares et al., 2014) and state derivative estimation (Bhasin et al., 2013; Kamalapurkar et al., 2014). Both of these methods suffer from the persistence of exitation(PE) condition that requires the state trajectory $\phi^{\hat{u}}(t;t_0,\zeta_0)$ to cover the entire operating domain for the convergence of the parameters to their optimal values. We have relaxed this condition where the integral technique is used in augmentation with the replay of the experience where every evaluation of the BE is intuitively formalized as a gained experience, and these experiences are kept in a history stack so that they can be iteratively used by the learning algorithm to improve data efficiency.

Therefore, to relax the PE condition, the we have developed a CL-based system identifier which is used to model the parametric estimate of the system drift dynamics and is used to simulate the experience by extrapolating the Bell Error (BE) over the unexplored territory in the operating domain thereby, prompting an exponential convergence of the parameters to their optimal values.

### 2.2.1 PARAMETRIC SYSTEM IDENTIFICATION

Defined by any compact set $C \subset \mathbb{R}$, the function $f$ can be defined using a neural network (NN) as $f(x) = \theta^T \sigma_f(Y^T x_1) + \epsilon_0(x)$ where, $x_1 = [1, x^T]^T \in \mathbb{R}^{n+1}$, $\theta \in \mathbb{R}^{n+1 X p}$ and $Y \in \mathbb{R}^{n+1 X p}$ indicates the constant unknown output-layer and hidden-layer NN weight, $\sigma_f : \mathbb{R}^p \to \mathbb{R}^{p+1}$ denotes a bounded NN activation function, $\epsilon_\theta: \mathbb{R}^n \to \mathbb{R}^n$ is the function reconstruction error, $p \in \mathbb{N}$ denotes the number of NN neurons. Using the universal functionional approximation property of single layer NNs, given a constant matrix $Y$ such that the rows of $\sigma_f(Y^T x_1)$ form a proper basis, there exist constant ideal weights $\theta$ and known constants $\bar{\theta}, \bar{\epsilon}_\theta, \bar{\epsilon}'_\theta \in \mathbb{R}$ such that $||\boldsymbol{\theta}|| \leq \bar{\theta} < \infty$, $sup_{x\epsilon C} ||\epsilon_\theta(x)|| \leq \bar{\epsilon}_\theta$, $sup_{x\epsilon C} ||\nabla_{\boldsymbol{x}}\epsilon_\theta(x)|| \leq \bar{\epsilon}_\theta$ where, $||.||$ denotes the Euclidean norm for vectors and the Frobenius norm for matrix (Lewis et al., 1998).

Taking into consideration an estimate $\hat{\theta} \in \mathbb{R}^{p+1 X n}$ of the weight matrix $\theta$, the function $f$ can be approximated by the function $\hat{f} : \mathbb{R}^{2n}$ X $\mathbb{R}^{p+1 X n} \to \mathbb{R}^n$ which is defined as $\hat{f}(\zeta, \hat{\theta}) = \hat{\theta}^T \sigma_\theta(\zeta)$, where $\sigma_\theta : \mathbb{R}^{2n} \to \mathbb{R}^{p+1}$ can be defined as $\sigma_\theta(\zeta) = \sigma_f(Y^T[1, e^T + \bar{x}^T]^T)$. An estimator for online identification of the drift dynamics is developed

$$\dot{\hat{x}} = \hat{\theta}^T \sigma_\theta(\zeta) + g(x)u + k\tilde{x} \qquad (9)$$

where, $\tilde{x} = x - \hat{x}$ and $k \,\epsilon\, \mathbb{R}$ R is a positive constant learning gain.

*Assumption 4:* A history stack containing recorded state-action pairs $\{x_j, u_j\}_{j=1}^{M}$ along with numerically computed state derivatives $\{\dot{\bar{x}}_j\}_{j=1}^{M}$ that satisfies $\lambda_{\min}\left(\sum_{j=1}^{M} \sigma_{fj}\sigma_{fj}^T\right) = \underline{\sigma_\theta} > 0$, $\|\dot{\bar{x}}_j - \dot{x}_j\| < \bar{d}, \forall j$ is available a priori, where $\sigma_{fj} \triangleq \sigma_f\left(Y^T\left[1, x_j^T\right]^T\right)$, $\bar{d} \in \mathbb{R}$ is a known positive constant, $\dot{x}_j = f(x_j) + g(x_j) u_j$ and $\lambda_{\min}(\cdot)$ denotes the minimum eigenvalue.

The weight estimates $\hat{\theta}$ are updated using the following CL based update law:

$$\dot{\hat{\theta}} = \Gamma_\theta \sigma_f\left(Y^T x_1\right)\tilde{x}^T + k_\theta \Gamma_\theta \sum_{j=1}^{M} \sigma_{fj}\left(\dot{\bar{x}}_j - g_j u_j - \hat{\theta}^T \sigma_{fj}\right)^T \tag{10}$$

where $k_\theta \in \mathbb{R}$ is a constant positive CL gain, and $\Gamma_\theta \in \mathbb{R}^{p+1 \times p+1}$ is a constant, diagonal, and positive definite adaptation gain matrix. Using the identifier, the BE in (3) can be approximated as

$$\hat{\delta}\left(\zeta, \hat{\theta}, \hat{W}_c, \hat{W}_a\right) = \bar{Q}(\zeta) + \hat{\mu}^T\left(\zeta, \hat{W}_a\right) R\hat{\mu}\left(\zeta, \hat{W}_a\right) \\ + \nabla_\zeta \hat{V}\left(\zeta, \hat{W}_a\right)\left(F_\theta(\zeta, \hat{\theta}) + F_1(\zeta) + G(\zeta)\hat{\mu}\left(\zeta, \hat{W}_a\right)\right) \tag{11}$$

The BE is now approximated as

$$\hat{\delta}\left(\zeta, \hat{\theta}, \hat{W}_c, \hat{W}_a\right) = \bar{Q}(\zeta) + \hat{\mu}^T\left(\zeta, \hat{W}_a\right) R\hat{\mu}\left(\zeta, \hat{W}_a\right) \\ + \nabla_\zeta \hat{V}\left(\zeta, \hat{W}_a\right)\left(F_\theta(\zeta, \hat{\theta}) + F_1(\zeta) + G(\zeta)\hat{\mu}\left(\zeta, \hat{W}_a\right)\right) \tag{12}$$

In equation (12), $F_\theta(\zeta, \hat{\theta}) = \begin{bmatrix} \hat{\theta}^T \sigma_\theta(\zeta) - g(x)g^+(x_d)\hat{\theta}^T \sigma_\theta\left(\begin{bmatrix} \mathbf{0}_{n \times 1} \\ x_d \end{bmatrix}\right) \\ \mathbf{0}_{n \times 1} \end{bmatrix}$, and $F_1(\zeta) = \left[\left(-h_d + g(e + x_d)g^+(x_d)h_d\right)^T, h_d^T\right]^T$.

### 2.2.2 VALUE FUNCTION APPROXIMATION

As $V^*$ and $\mu^*$ are functions of the state $\zeta$, the optimization problem as defined in Section 2.2 is quite an intractable one, so the optimal value function is now represented as $\mathcal{C} \subset \mathbb{R}^{2n}$ using a NN as $V^*(\zeta) = W^T \sigma(\zeta) + \epsilon(\zeta)$, where $W \in \mathbb{R}^L$ denotes a vector of unknown NN weights, $\sigma : \mathbb{R}^{2n} \to \mathbb{R}^L$ indicates a bounded NN activation function, $\epsilon : \mathbb{R}^{2n} \to \mathbb{R}$ defines the function reconstruction error, and $L \in \mathbb{N}$ denotes the number of NN neurons. Considering the universal function approximation property of single layer NNs, for any compact set $\mathcal{C} \subset \mathbb{R}^{2n}$, there exist constant ideal weights $W$ and known positive constants $\bar{W}, \bar{\epsilon}$, and $\bar{\epsilon}' \in \mathbb{R}$ such that $\|W\| \leq \bar{W} < \infty \sup_{\zeta \in \mathcal{C}} \|\epsilon(\zeta)\| \leq \bar{\epsilon}$, and $\sup_{\zeta \in \mathcal{C}} \|\nabla_\zeta \epsilon(\zeta)\| \leq \bar{\epsilon}'$ (Lewis et al., 1998).

A NN representation of the optimal policy is obtained as

$$\mu^*(\zeta) = -\frac{1}{2}R^{-1}G^T(\zeta)\left(\nabla_\zeta \sigma^T(\zeta)W + \nabla_\zeta \epsilon^T(\zeta)\right) \tag{13}$$

Taking the estimates $\hat{W}_c$ and $\hat{W}_a$ for the ideal weights $W$, the optimal value function and the optimal policy are approximated as $\hat{V}\left(\zeta, \hat{W}_c\right) = \hat{W}_c^T \sigma(\zeta)$, $\hat{\mu}\left(\zeta, \hat{W}_a\right) = -\frac{1}{2}R^{-1}G^T(\zeta)\nabla_\zeta \sigma^T(\zeta)\hat{W}_a$. The optimal control problem is therefore recast as to find a set of weights $\hat{W}_c$ and $\hat{W}_a$ online to minimize the error $\hat{E}_{\hat{\theta}}\left(\hat{W}_c, \hat{W}_a\right) = \sup_{\zeta \in \chi}\left|\hat{\delta}\left(\zeta, \hat{\theta}, \hat{W}_c, \hat{W}_a\right)\right|$ for a given $\hat{\theta}$, while simultaneously improving $\hat{\theta}$ using the CL-based update law and ensuring stability of the system using the control law

$$u = \hat{\mu}\left(\zeta, \hat{W}_a\right) + \hat{u}_d(\zeta, \hat{\theta}) \tag{14}$$

where, $\hat{u}_d(\zeta, \hat{\theta}) = g_d^+\left(h_d - \hat{\theta}^T \sigma_{\theta d}\right)$, and $\sigma_{\theta d} = \sigma_\theta\left(\begin{bmatrix} \mathbf{0}_{1 \times n} & x_d^T \end{bmatrix}^T\right)$. $\sigma_\theta\left(\begin{bmatrix} \mathbf{0}_{1 \times n} & x_d^T \end{bmatrix}^T\right)$. The error between $u_d$ and $\hat{u}_d$ is included in the stability analysis based on the fact that the error trajectories generated by the system $\dot{e} = f(x) + g(x)u - \dot{x}_d$ under the controller in (14) are identical to the error trajectories generated by the system $\dot{\zeta} = F(\zeta) + G(\zeta)\mu$ under the control law $\mu = \hat{\mu}\left(\zeta, \hat{W}_a\right) + g_d^+ \tilde{\theta}^T \sigma_{\theta d} + g_d^+ \epsilon_{\theta d}$, where $\epsilon_{\theta d} \triangleq \epsilon_\theta(x_d)$.

### 2.2.3 EXPERIENCE SIMULATION

The simulation of experience is implemented by minimizing a squared sum of BEs over finitely many points in the state space domain as the calculation of the extremum (supremum) in $\hat{E}_{\hat{\theta}}$ is not tractable. The details of the analysis has been explained in **Appendix A.2** which facilitates the aforementioned approximation.

### 2.2.4 STABILITY AND ROBUSTNESS ANALYSIS

To perform the stability analysis, we take the non-autonomous form of the value function (Kamalapurkar et al., 2015) defined by $V_t^* : \mathbb{R}^n \, \mathrm{X} \, \mathbb{R} \to \mathbb{R}$ which is defined as $V_t^*(e, t) = V^*\left( \left[ e^T, x_d^T(t) \right]^T \right), \forall e \in \mathbb{R}^n, t \in \mathbb{R}$, is positive definite and decrescent. Now, $V_t^*(0, t) = 0, \forall t \in \mathbb{R}$ and there exist class $\mathcal{K}$ functions $\underline{v} : \mathbb{R} \to \mathbb{R}$ and $\bar{v} : \mathbb{R} \to \mathbb{R}$ such that $\underline{v}(\|e\|) \leq V_t^*(e, t) \leq \bar{v}(\|e\|)$, for all $e \in \mathbb{R}^n$ and for all $t \in \mathbb{R}$. We take an augemented state given as $Z \in \mathbb{R}^{2n+2L+n(p+1)}$ is defined as

$$Z = \left[ \begin{array}{ccccc} e^T, & \tilde{W}_c^T, & \tilde{W}_a^T, & \tilde{x}^T, & (\mathrm{vec}(\tilde{\theta}))^T \end{array} \right]^T \tag{15}$$

and a candidate Lyapunov function is defined as

$$V_L(Z, t) = V_t^*(e, t) + \frac{1}{2}\tilde{W}_c^T \Gamma^{-1}\tilde{W}_c + \frac{1}{2}\tilde{W}_a^T \tilde{W}_a \frac{1}{2}\tilde{x}^T \tilde{x} + \frac{1}{2}\mathrm{tr}\left( \tilde{\theta}^T \Gamma_\theta^{-1}\tilde{\theta} \right) \tag{16}$$

where, vec $(\cdot)$ denotes the vectorization operator. From the weight update in **Appendix A.2** we get positive constants $\underline{\gamma}, \bar{\gamma} \in \mathbb{R}$ such that $\underline{\gamma} \leq \left\| \Gamma^{-1}(t) \right\| \leq \bar{\gamma}, \forall t \in \mathbb{R}$. Taking the bounds on $\Gamma$ and $V_t^*$ and the fact that $\mathrm{tr}\left( \tilde{\theta}^T \Gamma_\theta^{-1}\tilde{\theta} \right) = (\mathrm{vec}(\tilde{\theta}))^T \left( \Gamma_\theta^{-1} \otimes \mathbb{I}_{p+1} \right) (\mathrm{vec}(\tilde{\theta}))$ the candidate Lyapunov function be bounded as

$$\underline{v_l}(\|Z\|) \leq V_L(Z, t) \leq \bar{v_l}(\|Z\|) \tag{17}$$

for all $Z \in \mathbb{R}^{2n+2L+n(p+1)}$ and for all $t \in \mathbb{R}$, where $v_l : \mathbb{R} \to \mathbb{R}$ and $\overline{v_l} : \mathbb{R} \to \mathbb{R}$ are class $\mathcal{K}$ functions. Now, Using (1) and the fact that $V_t^*(e(t), t) = \dot{V}^*(\zeta(t)), \forall t \in \mathbb{R}$, the time-derivative of the candidate Lyapunov function is given by

$$\dot{V}_L = \nabla_\zeta V^* (F + G\mu^*) - \tilde{W}_c^T \Gamma^{-1}\dot{\tilde{W}}_c - \frac{1}{2}\tilde{W}_c^T \Gamma^{-1}\dot{\Gamma}\Gamma^{-1}\tilde{W}_c$$
$$- \tilde{W}_a^T \dot{\tilde{W}}_a + \dot{V}_0 + \nabla_\zeta V^* G\mu - \nabla_\zeta V^* G\mu^* \tag{18}$$

Under sufficient gain conditions (Kamalapurkar et al., 2014), using (9), (10)-(13), and the update laws given by $\hat{W}_c$, $\dot{\Gamma}$ and $\hat{W}_a$ the time-derivative of the candidate Lyapunov function can be bounded as

$$\dot{V}_L \leq -v_l(\|Z\|), \forall \|Z\| \geq v_l^{-1}(\iota), \forall Z \in \chi \tag{19}$$

where $\iota$ is a positive constant, and $\chi \subset \mathbb{R}^{2n+2L+n(p+1)}$ is a compact set. Considering (13) and (15), the theorem 4.18 in (Khalil., 2002) can be used to establish that every trajectory $Z(t)$ satisfying $\|Z(t_0)\| \leq \overline{v_l}^{-1}(v_l(\rho))$, where $\rho$ is a positive constant, is bounded for all $t \in \mathbb{R}$ and satisfies $\limsup_{t \to \infty} \|Z(t)\| \leq \underline{v_l}^{-1}\left( \overline{v_l}\left( v_l^{-1}(\iota) \right) \right)$. This aforementioned analysis addresses the stability issue of the closed loop system.

The robustness criterion requires the algorithm to satisfy the following inequality (Gao et al., 2014) in the presence of external disturbances with a pre-specified performance index $\gamma$ known as the H-infinity ($H_\infty$) performance index, given by

$$\int_0^t \|y(t)\|^2 dt < \gamma^2 \int_0^t \|w(t)\| dt \tag{20}$$

where, *y(t)* is the output of the system, *w(t)* is the factor that accounts for the modeling errors, parameter uncertainties and external disturbances and $\gamma$ is the ratio of the output energy to the disturbance in the system.

Using (1) and the fact that $V_t^*(e(t), t) = \dot{V}^*(\zeta(t)), \forall t \in \mathbb{R}$, the time-derivative of the candidate Lyapunov function is given by

$$\dot{V}_L = \nabla_\zeta V^* (F + G\mu^*) - \tilde{W}_c^T \Gamma^{-1}\dot{\tilde{W}}_c - \frac{1}{2}\tilde{W}_c^T \Gamma^{-1}\dot{\Gamma}\Gamma^{-1}\tilde{W}_c$$
$$- \tilde{W}_a^T \dot{\tilde{W}}_a + \dot{V}_0 + \nabla_\zeta V^* G\mu - \nabla_\zeta V^* G\mu^* \tag{21}$$

Gao et al. (2014) has shown if (22) and (23) is satisfied, then it can written that

$$0 < V_L(T) = \int_0^t \dot{V}_L(t) \leq -\int_0^t y^T(t)y(t)dt + \gamma^2 \int_0^t w^T(t)w(t)dt \qquad (22)$$

Thus, the performance inequality constraint given by $\int_0^t \|y(t)\|^2 dt < \gamma^2 \int_0^t \|w(t)\| dt$ in terms of $\gamma$ is satisfied.

## 3    SIMULATION RESULTS AND DISCUSSION

Here, we are going to present the simulation results to demonstrate the performance of the proposed method with the fuel management system of the hybrid electric vehicle. The proposed concurrent learning based RL optimization architecture has been shown in the Figure 1.

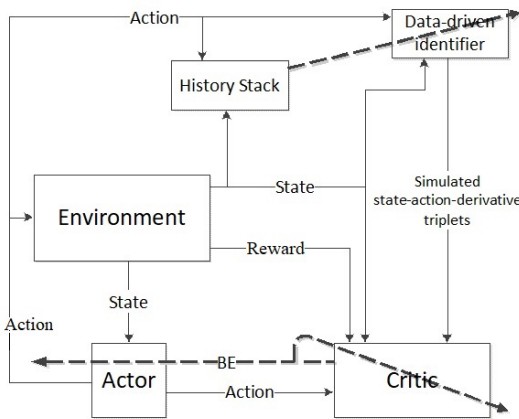

Figure 1: Reinforcement Learning-based Optimization Architecture

In this architecture, the simulated state-action-derivative triplets performs the action of concurrent learning to approximate the value function weight estimates to minimize the bell error (BE). The history stack is used to store the evaluation of the bell error which is carried out by a dynamic system identifier as a gained experience so that it can iteratively used to reduce the computational burden.

A simple two dimensional model of the fuel management system is being considered for the simulation purpose to provide a genralized solution that can be extended in other cases of high dimensional system.

We consider a two dimensional non-linear model given by

$$f = \begin{bmatrix} x_1 & x_2 & 0 & 0 \\ 0 & 0 & x_1 & x_2(1 - (cos(2x_1 + 2)^2)) \end{bmatrix} * \begin{bmatrix} a \\ b \\ c \\ d \end{bmatrix}, \ g = \begin{bmatrix} 0 \\ cos(2x_1 + 2) \end{bmatrix}, \ w(t) = sin(t)$$

$$(23)$$

where $a, b, c, d \in \mathbb{R}$ are unknown positive parameters whose values are selected as $a = -1$, $b = 1$, $c = -0.5$, $d = -0.5$, $x_1$ and $x_2$ are the two states of the hybrid electric vehicle given by the charge present in the battery and the amount of fuel in the car respectively and $w(t) = sin(t)$ is a sinusoidal disturbance that is used to model the external disturbance function. The control objective is to minimize the cost function given by $J(\zeta, \mu) = \int_0^t r(\zeta(\tau), \mu(\tau))d\tau$ where, the local cost $r : \mathbb{R}^{2n} X R^m \to \mathbb{R}$ is given as $r(\zeta, \tau) = Q(e) + \mu^T R\mu$, $R \in \mathbb{R}^{mXm}$ is a positive definite symmetric matrix and $Q : \mathbb{R}^n \to \mathbb{R}$ is a continous positive definite function, while following the desired trajectory $\bar{x}$ We chhose $Q = I_{2x2}$ and $R = 1$. The optimal value function and optimal control for the system (15) are $V^*(x) = \frac{1}{2}x_1^2 + \frac{1}{2}x_2^2$ and $u^*(x) = -cos(2(x_1) + 2)x_2$. The basis

function $\sigma : \mathbb{R}^2 \rightarrow \mathbb{R}^3$ for value function approximation is $\sigma = [x_1^2, \ x_1^2 x_2^2, \ x_2^2]$. The ideal weights are $W = [0.5, 0, 1]$. The initial value of the policy and the value function weight estimates are $\hat{W}_c$ = $\hat{W}_a = [1, 1, 1]^T$ , least square gain is $\Gamma(0) = 100 I_{3X3}$ and that of the system states are $x(0)$ = $[-1, -1]^T$. The state estimates $\hat{x}$ and $\hat{\theta}$ are initialized to 0 and 1 respectively while the history stack for the CL is updated online. Here, Figure 2 and Figure 3 shows the state trajectories obtained by the

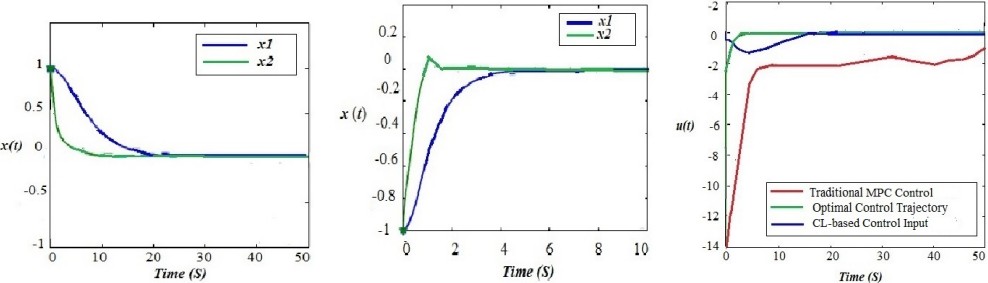

Figure 2: State Trajectories  Figure 3: State Trajectories  Figure 4: Control Input

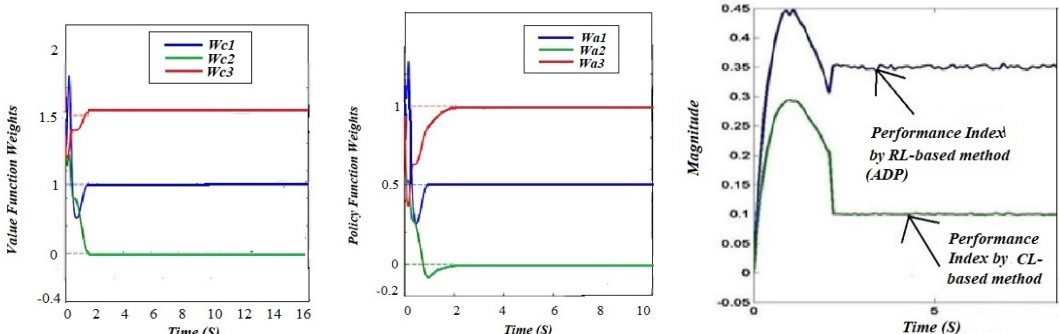

Figure 5: Value Function  Figure 6: Policy Function  Figure 7: Performance Index

traditional RL methods and that obtained by the CL-based RL optimization technique respectively in the presence of disturbances. It can be stated that settling time of trajectories obtained by the proposed method is significantly less (almost 40 percent) as compared with that of the conventional RL strategies thus justifying the uniqueness of the method and causing a saving in fuel consumption by about 40-45 percent. Figure 4 shows the corresponding control inputs whereas Figure 5 and Figure 6 indicates the convergence of the NN weight functions to their optimal values. The $H_\infty$ performance index in Figure 7 shows a value of 0.3 for the CL-based RL method in comparison to 0.45 for the traditional RL-based control design which clearly establishes the robustness of our proposed design.

## 4 CONCLUSION

In this paper, we have proposed a robust concurrent learning based deep Rl optimization strategy for hybrid electric vehicles. The uniqueness of this method lies in use of a concurrent learning based RL optimization strategy that reduces the computational complexity significanty in comparison to the traditional RL approaches used for the fuel management system mentioned in the literature. Also, the use of the the H-infinity ($H_\infty$) performance index in case of RL optimization for the first time takes care of the robustness problems that most the fuel optimization nethods suffer from. The simulation results validate the efficacy of the method over the conventional PID, MPC as well as traditional RL based optimization techniques. Future work will generalize the approach for large-scale partially observed uncertain systems and it will also incorporate the movement of neighbouring RL agents.

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

# A APPENDIX

## A.1 THE GRADIENT DESCENT ALGORITHM

The Gradient Descent Algorithm has been explained as follows:

***Algorithm; Gradient Descent***

**Input :** Design Parameters $U^{(0)} = u_t^0,\ \alpha, h\ \epsilon\ \mathbb{R}$

**Output :** Optimal control sequence $\{\bar{u}_t\}$

**1**. $n \leftarrow 0, \nabla_U J_d \left( x_0, U^{(0)} \right) \leftarrow \epsilon$

**2**. **while** $\nabla_U J_d \left( x_0, U^{(n)} \right) \geq \epsilon$ **do**

**3**. Evaluate the cost function with control $U^{(n)}$

**4**. Perturb each control variable $u_i^{(n)}$ by $h$, $i = 0, \cdots, t$, and calculate the gradient vector $\nabla_U J_d \left( x_0, U^{(n)} \right)$ using (7) and (8)

**5**. Update the control policy: $U^{(n+1)} \leftarrow U^{(n)} - \alpha \nabla_U J_d \left( x_0, U^{(n)} \right)$

**6**. $n \leftarrow n + 1$

**7**. **end**

**8**. $\{\bar{u}_t\} \leftarrow U^{(n)}$

## A.2 EXPERIENCE SIMULATION

*Assumption 5:* (Kamalapurkar et al., 2014) There exists a finite set of points $\{\zeta_i \in \mathcal{C} \mid i = 1, \cdots, N\}$ and a constant $\underline{c} \in \mathbb{R}$ such that $0 < \underline{c} = \frac{1}{N} \left( \inf_{t \in \mathbb{R}_{\geq t_0}} \left( \lambda_{\min} \left\{ \sum_{i=1}^N \frac{\omega_i \omega_i^T}{\rho_i} \right\} \right) \right)$ where $\rho_i = 1 + \nu \omega_i^T \Gamma \omega_i \in \mathbb{R}$, and $\omega_i = \nabla_\zeta \sigma \left( \zeta_i \right) \left( F_\theta \left( \zeta_i, \hat{\theta} \right) + F_1 \left( \zeta_i \right) + G \left( \zeta_i \right) \hat{\mu} \left( \zeta_i, \hat{W}_a \right) \right)$.

Using *Assumption 5*, simulation of experience is implemented by the weight update laws given by

$$\hat{W}_c = -\eta_{c1} \Gamma \frac{\omega}{\rho} \hat{\delta}_t - \frac{\eta_{c2}}{N} \Gamma \sum_{i=1}^N \frac{\omega_i}{\rho_i} \hat{\delta}_{ti} \tag{24}$$

$$\dot{\Gamma} = \left( \beta \Gamma - \eta_{c1} \Gamma \frac{\omega \omega^T}{\rho^2} \Gamma \right) \mathbf{1}_{\{\|\Gamma\| \leq \bar{\Gamma}\}}, \|\Gamma \left( t_0 \right)\| \leq \bar{\Gamma}, \tag{25}$$

$$\dot{\hat{W}}_a = -\eta_{a1} \left( \hat{W}_a - \hat{W}_c \right) - \eta_{a2} \hat{W}_a + \left( \frac{\eta_{c1} G_\sigma^T \hat{W}_a \omega^T}{4\rho} + \sum_{i=1}^N \frac{\eta_{c2} G_{\sigma i}^T \hat{W}_a \omega_i^T}{4N \rho_i} \right) \hat{W}_c \tag{26}$$

where, $\omega = \nabla_\zeta \sigma(\zeta) \left( F_\theta(\zeta, \hat{\theta}) + F_1(\zeta) + G(\zeta)\hat{\mu}\left(\zeta, \hat{W}_a\right) \right)$, $\Gamma \in \mathbb{R}^{L \times L}$ is the least-squares gain matrix, $\bar{\Gamma} \in \mathbb{R}$ denotes a positive saturation constant, $\beta \in \mathbb{R}$ indicates a constant forgetting factor, $\eta_{c1}, \eta_{c2}, \eta_{a1}, \eta_{a2} \in \mathbb{R}$ defines constant positive adaptation gains, $1_{\{\cdot\}}$ denotes the indicator function of the set $\{\cdot\}$, $G_\sigma = \nabla_\zeta \sigma(\zeta) G(\zeta) R^{-1} G^T(\zeta) \nabla_\zeta \sigma^T(\zeta)$, and $\rho = 1 + \nu \omega^T \Gamma \omega$, where $\nu \in \mathbb{R}$ is a positive normalization constant. In the above weight update laws, for any function $\xi(\zeta, \cdot)$, the notation $\xi_i$, is defined as $\xi_i = \xi(\zeta_i, \cdot)$, and the instantaneous BEs $\hat{\delta}_t$ and $\hat{\delta}_{ti}$ are given as $\hat{\delta}_t = \hat{\delta}\left(\zeta, \hat{W}_c, \hat{W}_a, \hat{\theta}\right)$ and $\hat{\delta}_{ti} = \hat{\delta}\left(\zeta_i, \hat{W}_c, \hat{W}_a, \hat{\theta}\right)$.

