# OpenReview forum: "A Robust Fuel Optimization Strategy For Hybrid Electric Vehicles: A Deep Reinforcement Learning Based Continuous Time Design Approach"
_ICLR.cc/2021/Conference — Reject_

### Official Review · AnonReviewer1 · 2020-10-28
**Lacks clarity to judge the paper objectively**

**Rating:** 3
**Confidence:** 4

**Review:**



### Summary
The paper proposed to use RL methods for control for hybrid vehicles fuel
strategy. The organisation of the paper is difficult to follow and I may have
missed some of the arguments the authors are making. The experimental section is
very thin with only a 2 d system simulation.

### To improve

1. How is this even possible ? "We now solve the open loop optimization problem
   using a general non-linear programming solver without actually knowing the
   exact form of the underlying dynamics" The cost doesn't have the dynamical system
   equation but you need to enforce it either as a constraint or solve is via
   typical single shooting methods. Constraint is mentioned in the section 2.1 but it
   is still incorrect to says that gradient descent is without access to dynamics.
   Do you mean to say that we learn dynamics from the data ? and model is
   assumed to be unknown at the design time ?
2. MPC and PID are not the same and I am not sure why they are clubbed together
   as a baseline. Especially in the experiment shown the system is simple
   enough to be learned by wide enough NN. Then we can use non-linear MPC with
   out of the box optimisers as SOTA baseline  for MPC ?
3. Authors learn a NN based model as part of the proposed method, MPC would be
   an algorithm that can use this model to optimise the control.
4. If the hybrid system model is as described in the experimental section I am
   struggling to see how this scholarship helps compared to standard optimal
   control methods ?
5. I found the overall paper very challenging to follow. There are large logical
   jumps. I am open to changing my review, if authors can show the clear benefits of their proposal.

### citations

1. correct source for DDP is Mayne [1]

### Language

1. This has no impact on technical rating of the paper and it does not directly
   contribute to the review score.
2. I had started to write all the typos and grammatical errors in the paper and
   I stopped as there are far too many of them. Please review this for language
   errors as this breaks flow of reading.

### Ref

1. Jacobson, D. H., & Mayne, D. Q. (1970). Differential Dynamic Programming. American Elsevier Publishing Company. https://books.google.co.uk/books?id=tA-oAAAAIAAJ

---

> ### Author Response · Authors · 2020-11-21
> **Addressing the reviewer's concerns about the submission on - A Robust Fuel Optimization Strategy For Hybrid Electric Vehicles: A Deep Reinforcement Learning Based Continuous Time Design Approach                                                                                                                                                      oach**
>
> First of all, we would like to thank the reviewer for his valuable comments on our work. This is absolutely essential to find out the flaws and would help us to come up with a much stronger technical explanation.
>
> C1: How is this even possible? "We now solve the open loop optimization problem using a general non-linear programming solver without actually knowing the exact form of the underlying dynamics" The cost doesn't have the dynamical system …… Do you mean to say that we learn dynamics from the data? and model is assumed to be unknown at the design time?
>
> A1: The open loop optimization problem is solved using a black box model of the non-linear dynamics, with sequence of input-output tests. The model dynamics is data driven and is unknown at the design phase. The model is learnt from the set of input-output tests. After we have learnt the model from the data, Gradient Descent is used to determine the nominal optimal state trajectory.
>
> C2: MPC and PID are not the same…….non-linear MPC with out of the box optimizers as SOTA baseline for MPC?
>
> A2: MPC and PID are not clubbed together, both the methods have been applied separately to the simulation model of a fuel management system of a hybrid electric vehicle. We have also tried Non-linear MPC with SOTA baseline, like many existing literatures, in this simulation model but it gives very poor results when encountering disturbances (Robustness)  as the value of the performance metric (H-infinity ($H_{\infty})$ performance index which is the ratio of the output energy to the control energy) for Non-linear MPC is 0.45 as compared to our method which has a value of 0.3.  It is being updated in our revised version (section 3). The analysis has been described in section 2.2.4. The analysis has been added in section 2.2.4.
>
> C3: Authors learn a NN based model as part of the proposed method, MPC would be an algorithm that can use this model to optimize the control.
>
> A3: Yes, MPC can be used but as already mentioned, it does not provide a robust optimal solution for fuel management of hybrid electric vehicles, which is one of the major focus and novelty of our work.
>
> C4: If the hybrid system model is as described in the experimental section……. compared to standard optimal control methods?
>
> A4: As already mentioned, our main focus is to provide a robust , computationally tractable optimal solution using RL approach for fuel optimization in hybrid electric vehicles. The motivation behind choosing this particular hybrid model is that it incorporates a  disturbance function to simulate the effects of disturbances that an original hybrid electric vehicle has to encounter. Our proposed algorithm gives an optimal solution even in the presence of disturbances as demonstrated by the experimental results which the traditional RL approaches (ADP, DDP, DQN) or other conventional like MPC, PID which are widely used in most of existing literatures for fuel optimization, fails to do (section 3). This has been modified and we will include this in our updated version.
>
> C5: …… if authors can show the clear benefits of their proposal.
>
> A5: The two major benefits of our proposal are:
>
> Development of a robust and computationally tractable optimal solution for fuel management problem of hybrid electric vehicles.
>
> a. Most of the SOTA RL approaches for fuel optimization in hybrid electric vehicles suffers from the problem of providing a sub-optimal solution in the presence of disturbances as demonstrated by the high settling time of state trajectories (23 -24 senonds) and high value of the H-infinity ($H_{\infty})$ performance index (0.45), in section 3. Our proposed algorithm reduces the settling time to about 5 seconds and the H-infinity ($H_{\infty})$ performance index has a value of 0.3 which clearly indicates that our approach is much more robust in comparison the SOTA RL and other conventional methods.  Our method also causes a fuel saving of about 45% more than the existing methods (section 3).
>
> b. The computational complexity of the proposed method is significantly less than the existing RL methods of fuel optimization in hybrid electric vehicles. As studied in the literature, the SOTA RL methods employ an actor-critic methods which generally visits most of the states in state action space to determine the value function and the corresponding action. however here we have the concept of concurrent learning for the first time in case of RL methods for fuel optimization. This greatly reduces the computational cost and overcomes the problem of existing methods which are infeasible to use in a high dimensional system like hybrid electric vehicles.
> This has been explained in detail in section 2 and 2.2.1 in our updated version. Thus this approach will be much cheaper and easier to implement and execute.
>
> A6: Mayne as a correct source for DDP has been mentioned in the references.
>
> A7: The typos and the grammatical errors have mostly been edited.

---

### Official Review · AnonReviewer4 · 2020-10-28
**Official Blind Review #4**

**Rating:** 5
**Confidence:** 3

**Review:**

This paper proposes a reinforcement learning framework for the fuel optimization problem in hybrid electric vehicles.

Strong points:
+ The design of the deep RL-based controller has been discussed in detail.
+ The simulation results show both the value and policy functions can quickly converge to their optimal values.
+ The $H_\infty$ performance index shows the RL-based method is more robust than the classical PID/MPC-based methods.

Weak points:
- The HEV fuel/energy functions are generally nonlinear non-convex functions that have been widely modeled in the past literature. It’s not clear how the authors consider the cost function as a quadratic formulation (Page 4 Section 2.1) and relate it to optimizing the HEV fuel consumption.
- The quality and completeness of Figure 1 needs to be improved.
- More details should be provided on how the data-driven identifiers work and interact with the actor-critic networks in Figure 1. What are the benefits of the proposed method, when compared with directly applying a commonly known actor-critic RL method such as DDPG or PPO to solving the problem?
- The estimated percentage of fuel savings with the proposed RL-based method over the benchmark studies (PID/MPC-based methods and optimal control-based method) is not discussed.
- For the two-dimensional nonlinear model given by equation 11, can this model be extended to consider other surrounding agents’ movement?

---

> ### Author Response · Authors · 2020-11-22
> **Addressing the reviewer's questions on the submission**
>
> C1: The HEV fuel/energy functions are generally nonlinear non-convex functions ………..fuel consumption.
>
> A1: A non-convex function can be represented by piecewise convex functions which can be used to approximate the original non-convex function globally. The piecewise convex function for fuel consumption have been used to formulate a quadratic cost function which is also convex.  This has been modified in our updated version
>
> C2: The quality and completeness of Figure 1 needs to be improved.
>
> A2: We have modified and improved it in our edited submission.
>
> C3: More details should be provided on ……. DDPG or PPO to solving the problem?
>
> A3: The details have been added in the simulation results and discussion section (section 4) to indicate the interaction of the data driven identifier with the CL based Actor-Critic Network.
>
> The major benefits of the proposed method as compared to a commonly known actor-critic RL mthod like DDPG or PPO are as follows:
> Development of a robust and computationally tractable optimal solution for fuel management problem of hybrid electric vehicles.
>
> a. Most of the SOTA RL approaches like DDPG and PPO for fuel optimization in hybrid electric vehicles suffers from the problem of providing a sub-optimal solution in the presence of disturbances as demonstrated by the high settling time of state trajectories (23 -24 senonds) and high value of the H-infinity ($H_{\infty})$ performance index (0.45), in section 3. Our proposed algorithm reduces the settling time to about 5 seconds and the H-infinity ($H_{\infty})$ performance index has a value of 0.3 which clearly indicates that our approach is much more robust in comparison the SOTA RL and other conventional methods. Our method also causes a fuel saving of about 45% more than the existing methods (section 3). The analysis has been added in section 2.2.4.
>
> b. The computational complexity of the proposed method is significantly less than the existing RL methods of fuel optimization in hybrid electric vehicles. As studied in the literature, the SOTA RL methods employ an actor-critic strategy (like DDPG or PPO) which generally visits most of the states in state action space to determine the value function and the corresponding action. however here we have used the concept of concurrent learning for the first time in case of RL methods for fuel optimization. This greatly reduces the computational cost and overcomes the problem of existing methods which are infeasible to use in a high dimensional system like hybrid electric vehicles. This has been explained in detail in section 2 and 2.2.1 in our updated version. Thus this approach will be much cheaper and easier to implement and execute.
>
> C4: The estimated percentage fuel savings …….not mentioned.
>
> A4: A good point to address, we have modified and added it in our updated version.
>
> C5: For the two-dimensional nonlinear…….  other surrounding agents’ movement?
>
> A5: Yes, the model can be extended to consider other agent’s movement and that is the future direction in which we plan to implement our algorithm in a manner so that it will able to communicate with other near by agents and take possible decisions.

---

### Official Review · AnonReviewer3 · 2020-10-29
**Official Blind Review**

**Rating:** 4
**Confidence:** 2

**Review:**

This work proposes a deep reinforcement learning-based optimization strategy to the fuel optimization problem for the hybrid electric vehicle. The problem has been formulated as a fully observed stochastic Markov Decision Process (MDP). A deep neural network is used to parameterize the policy and value function. A continuous time representation of the problem  is also used compared to conventional techniques which mostly use a discrete time formulation.

The paper is very convoluted, with lots of different novelties, making it hard to understand the main contributions and how much each of the contributions truly affect the final results. Due to this, experiments are also lacking, testing the different aspects of the proposed approach. I.e., is the proposed reward model truly necessary? Does the overall RL model have to be so complex for this to work well? Perhaps a simpler model would work just as well or even better? What motivates the use of each of the different aspects of the model and each of the assumptions that are made? For example, the paper mentions "the convergence of the traditional RL requires sufficient exploration of the state-action space" but fails to explain how the proposed RL approach addresses this issue. In fact the proposed approach seems to be a straightforward application of actor-critic methods. The experiments seems the core contribution but there is only one page and isn't able to be replicated and thus, difficult to really evaluate. As an applied work, the solution is very specialized such that I am unsure that the observations would transfer to another domain.

---

> ### Author Response · Authors · 2020-11-22
> **Addressing the reviewer's questions on the submission**
>
> C1: ........main contributions and how much each of the contributions truly affect the final results.
>
> A1: The two major contributions of this work are development of a 1. Robust and 2. Computationally tractable optimal solution for fuel management of hybrid electric vehicles.
>
> From the experimental results. it can be understood that the proposed approach results in much settling time of the state trajectories (5 secs) and a lower value of the H-infinity ($H_{\infty})$ performance index  (0.3) as compared to the SOTA RL approaches like ADP, DDP and PPO which gives a settling time of 23-24 seconds for the state trajectories and value of the H-infinity ($H_{\infty})$ performance index as 0.45. This is a core contribution of the work because we have been able to formulate a robust optimal solution for fuel management system of hybrid electric vehicles which has to encounter a lot of different kinds of disturbances and noise during its operation. The SOTA RL algorithms provide only a sub-optimal solution in the presence of these disturbances as demonstrated by the results. Also the amount is fuel savings is around 45% more than the traditional approaches. The analysis has been added in section 2.2.4.
>
> The computational complexity of this method is very less than the traditional RL methods used for fuel optimization problem as described in section 2 and 2.2.1. Thus it results in a much cheaper and efficient implementation of the method.
>
> C2: experiments are also lacking, testing the different aspects of the proposed approach. .......Perhaps a simpler model would work just as well or even better?
>
> A2: Several experiments have been carried out to test the effects of the different approaches on fuel optimization problem of hybrid electric vehicles. From those experiments, we have concluded that the traditional SOTA RL approaches are not being robust  as shown in the experimental results. This has motivated us to come up a scheme that would be able to provide a robust optimal solution. The conventional methods like PID, MPC are also not able to handle the disturbances having a varying range of frequencies that a hybrid electric vehicle has to encounter.
>
> We have considered this reward model because we have tested it and seen that it is being able to reduce the computational complexity of the simpler models that the reviewer has mentioned. Because a hybrid electric vehicle is very high dimensional system so we need an algorithm that would be able to address this issue. The simpler SOTA RL approaches have failed to do that.
>
> Also, this model has been able to give a robust solution. So these two properties has motivated us to develop this model and algorithm.
>
> C3: For example, the paper mentions "the convergence of the traditional RL requires sufficient exploration of the state-action space" but fails to explain how the proposed RL approach addresses this issue. In fact the proposed approach seems to be a straightforward application of actor-critic methods.
>
> A3: The proposed RL approach has used the concept of concurrent learning to overcome the issue of computational complexity ("the convergence of the traditional RL requires sufficient exploration of the state-action space") as described in section 2.2 and 2.2.1. The concurrent learning overcomes the problem of computational complexity in the manner that it uses a history stack to store the evaluation of the Bellman Error as a gained experience from the previous states that it has already visited and simultaneously carrying out an extrapolation of the Bellman Error over the unexplored territory of the state-action space using a dynamic system identifier. Thus, we do not need to visit every state which is required for a good parametric approximation of the Value Function. This estimation is required for convergence of the parameters to their optimal values that is important for the stability of the closed loop system. This is how it address this issue.
>
> C4: As an applied work, the solution is very specialized such that I am unsure that the observations would transfer to another domain.
>
> A4: Although the solution is meant for fuel management of hybrid electric vehicles but this method can be generalized so that it can be applied to any high dimensional practical system plagued by a wide range of disturbances. It will provide a computationally tractable and robust optimal solution.

---

### Official Review · AnonReviewer2 · 2020-11-01
**Little original contribution and not really about fuel optimization**

**Rating:** 2
**Confidence:** 3

**Review:**

This paper purports to be about Hybrid vehicle fuel optimization using RL. In reality, most of the content seems to be a long derivation of a continuous-space generic RL based controller for a trajectory optimization problem. There does not seem to be much that is  novel in this entire presentation and if there is, the authors have not explicated it in the introduction or other sections. In the last section a very brief simulation-based experiment is described supporting the hybrid vehicle part. The bulk of the paper derives a controller using what seems to me to be standard ideas or minor variations thereof. Authors or other reviewers can correct me if I'm wrong but if so, the sheer density of  sec 2, esp 2.2 esp. made it hard to assess this (and I lacked motivation given what I perceived about the overall paper structure), so I would still argue for reject on clarity grounds.

The introduction has a comprehensive literature review of methods that have been used for open loop trajectory control, fuel optimization and so on.  This should be appreciated.

Coming to the claims in the introduction: First we solve an open loop trajectory optimization problem, then design an RL controller to track the nominal trajectory. There is nothing new in this idea. The second point brings up the use of something called Concurrent Learning as a contribution which is mentioned exactly once again in sec 2.1 in passing and never defined or referred to again. Perhaps 2.2 onwards describes a use of it but I can't tell.  Contribution #3 seems to be that using H-\infinity as a performance measure for an RL control algorithm is proposed for the first time, but again no description or justification is given.

Sec 2: up to 2.1 seems pretty straightforward.

2.2: Assumption 2: is g^+ the usual psuedo-inverse? It is usually defined with an extra A^T at the end.  is there a typo in assumption 3?  from that condition it seems that gg^+ should be identity, i dont think this is intended. Where is novelty in the rest of this section? Using HJB etc is standard. The description of various approximation schemes is ok but doesnt lead anywhere, unless 2.2.1-2 represents Concurrent learning.  sec 2.2.1 just seems to say we can approximate the dynamics using an NN. sec 2.2.2 approximates the value function using a NN as well. The universal function approximation theorem etc is invoked, but what new thing is really being said about VF approximation? I couldn't tell. What is the CL-based update law?

2.2.4: I dont remember enough about lyapunov conditions etc to dive deeply into this.  I know that lyapunov conditions are used to show stability of controllers. Is this a straightforward application of this method then?

sec 3: Since the main thrust of the paper is fuel optimization, absolutely no explanation or citation is given for why these particular system dynamics are relevant. Indeed, they seem far simpler than I would expect such a complex system to have, but I am definitely no expert. The results seem fine but not enough to justify this paper for ICLR as a deep and original contribution on learning representations.

---

> ### Author Response · Authors · 2020-11-22
> **Addressing the reviewer's questions on the submission**
>
> C1: There does not seem to be much that is novel in this entire presentation and if there is, the authors have not explicated it in the introduction ......... .
>
> A1: The novelties in this presentation are development of 1. Robust and 2. Computationally tractable optimal solution for fuel management system of hybrid electric vehicles using concurrent learning based approach.
>
> We have modified and added the description in section 2 and 2.2.1 for the computationally tractable solution analysis and section 3 for robustness analysis.
>
> Most of the SOTA RL approaches used for fuel optimization like ADP, DDP, PPO are computationally quite inefficient because they require the agent to visit every states to determine the value function and the corresponding action.  However, for a high dimensional system like hybrid electric vehicle, this approach fails to address the issue and therefore motives us to come up with design that solves the problem in an efficient manner.
>
> The proposed RL approach has used the concept of concurrent learning to overcome the issue of computational complexity ("the convergence of the traditional RL requires sufficient exploration of the state-action space") as described in section 2.2 and 2.2.1. The concurrent learning overcomes the problem of computational complexity in the manner that it uses a history stack to store the evaluation of the Bellman Error as a gained experience from the previous states that it has already visited and simultaneously carrying out an extrapolation of the Bellman Error over the unexplored territory of the state-action space using a dynamic system identifier. Thus, we do not need to visit every state which is required for a good parametric approximation of the Value Function. This estimation is required for convergence of the parameters to their optimal values that is important for the stability of the closed loop system. This is how it address this issue.
>
> Another novel feature of our method is that it gives a robust optimal solution in the presence of disturbances. The traditional RL approaches like DDPG and PPO for fuel optimization in hybrid electric vehicles suffers from the problem of providing a sub-optimal solution in the presence of disturbances as demonstrated by the high settling time of state trajectories (23 -24 senonds) and high value of the H-infinity ( performance index (0.45), in section 3. Our proposed algorithm reduces the settling time to about 5 seconds and the H-infinity ( performance index has a value of 0.3 which clearly indicates that our approach is much more robust in comparison the SOTA RL and other conventional methods. Our method also causes a fuel saving of about 45% more than the existing methods (section 3).
>
> C2: The second point brings up the .........for the first time, but again no description or justification is given.
>
> A2: The justification and the explanation has been added in the sections 2, 2.2.1 and 2.2.2 for the computational tractability analysis and section 2.2.4 for the robustness analysis.
>
> C3:  Where is novelty in the rest of this section? Using HJB etc is standard. The description of various approximation schemes is ok but doesnt lead anywhere, unless 2.2.1-2 represents Concurrent learning. sec 2.2.1............What is the CL-based update law?
>
> A3: The novelty is the use of Concurrent Learning based RL approaches for the first time in fuel management problem of hybrid electric vehicles.  The value function approximation has been done using concurrent learning in 2.2. The cl-based update law has been mentioned in section 2.1.
>
> C4: I dont remember enough about lyapunov conditions etc to dive deeply into this. I know that lyapunov conditions are used to show stability of controllers. Is this a straightforward application of this method then?
>
> A4: The lyapunov based stability analysis has been carried out to show that our approach is stable in comparison to ther CL-based approaches whose stability analysis is quite intarctable when considering the estimation error. We have taken estimation error in our calculation and have shown that the algorithm is stable which is quite a significant improvement over the exsting CL based approaches.
>
> A5:  As already mentioned, our main focus is to provide a robust , computationally tractable optimal solution using RL approach for fuel optimization in hybrid electric vehicles. The motivation behind choosing this particular hybrid model is that it incorporates a disturbance function to simulate the effects of disturbances that an original hybrid electric vehicle has to encounter. Our proposed algorithm gives an optimal solution even in the presence of disturbances as demonstrated by the experimental results which the traditional RL approaches (ADP, DDP, DQN) or other conventional like MPC, PID which are widely used in most of existing literatures for fuel optimization, fails to do (section 3). This has been modified and we will include this in our updated version.

---

### Author Response · Authors · 2020-11-24
**A Robust Fuel Optimization Strategy For Hybrid Electric Vehicles: A Deep Reinforcement Learning Based Continuous Time Design Approach**

Dear Reviewers,

Thank you for your time and effort in going through our paper and pointing out the faults, we have been able to address most of the issues that have been raised by the reviewers. However, just to give an idea, our main focus is to develop a robust and computationally tractable optimal solution for fuel optimization problem of hybrid electric vehicles. Therefore, we have first developed a concurrent-learning based robust RL fuel optimization strategy for hybrid electric vehicles and then have developed a simulation model where we have carried out experiments with traditional RL approaches and also MPC methods and compared the results with our proposed scheme to validate the efficacy of design scheme.

We have uploaded our rebuttal version.

Sincerely,

The authors.

---

### Decision · Program_Chairs · 2021-01-07
**Final Decision**

**Decision:**

Reject

**Comment:**

This paper proposes to solve the fuel optimization problem in hybrid electric vehicles using reinforcement learning. The work is interesting, but the reviewers consider it lacks novelty and it has different concerns on the assumptions of the modeling. The paper is quite difficult to follow.